# Comparative Evaluation of Three Commercial Quantitative Real-Time PCRs Used in Japan for Bovine Leukemia Virus

**DOI:** 10.3390/v14061182

**Published:** 2022-05-28

**Authors:** Syuji Yoneyama, Sota Kobayashi, Towa Matsunaga, Kaoru Tonosaki, Dongze Leng, Yusuke Sakai, Shinji Yamada, Atsushi Kimura, Toshihiro Ichijo, Hirokazu Hikono, Kenji Murakami

**Affiliations:** 1Graduate School of Veterinary Sciences, Iwate University, Morioka 020-8550, Japan; a3118010@iwate-u.ac.jp (S.Y.); a3118012@iwate-u.ac.jp (D.L.); sakai-yusuke@zenoaq.jp (Y.S.); yamadas@iwate-u.ac.jp (S.Y.); 2Division of Zoonosis Research, National Institute of Animal Health, National Agriculture and Food Research Organization, Tsukuba 305-0856, Japan; sotaco@affrc.go.jp; 3Department of Veterinary Sciences, Faculty of Agriculture, Iwate University, Morioka 020-8550, Japan; a8617026@iwate-u.ac.jp (T.M.); akimura@iwate-u.ac.jp (A.K.); ichijo@iwate-u.ac.jp (T.I.); 4Department of Plant Biosciences, Faculty of Agriculture, Iwate University, Morioka 020-8550, Japan; tonosaki@yokohama-cu.ac.jp; 5Department of Animal Sciences, Teikyo University of Science, Tokyo 120-0045, Japan; hhikono@ntu.ac.jp

**Keywords:** bovine leukemia virus, BLV, commercial kit, provirus load, qPCR

## Abstract

Bovine leukemia virus (BLV) is an oncogenic virus belonging to the genus *Deltaretrovirus* and is the causative agent of enzootic bovine leukosis. Proviral load (PVL) determined by real-time quantitative PCR (qPCR) is now widely used as an indicator of not only BLV infection, but also BLV disease progression. To interpret PVLs determined by different qPCRs used in Japan, we compared a chimeric cycling probe-based qPCR, CY415, targeting the BLV tax region; a TaqMan probe-based qPCR, RC202, targeting the BLV pol region; and a TaqMan probe-based qPCR, CoCoMo, targeting the BLV long terminal repeat (LTR) region. Whole-blood samples collected from 317 naturally BLV-infected cattle (165 Holstein–Friesian and 152 Japanese Black) and tumor tissue samples collected from 32 cattle at a meat inspection center were used. The PVLs determined by each qPCR were strongly correlated. However, the PVL and the proportion of BLV-infected cells determined by RC202 or CoCoMo were significantly higher than those determined by CY415. Genetic analysis of three tumor tissue samples revealed that LTR region mutations or a deletion affected the PVL determined by CoCoMo. These results suggest that the TaqMan-based RC202 or CoCoMo qPCR is better than CY415 for BLV PVL analysis. However, qPCR target region mutations were not rare in tumors and could hamper PVL analysis by using qPCR.

## 1. Introduction

Bovine leukemia virus (BLV) is an RNA virus belonging to the genus *Deltaretrovirus,* family *Retroviridae*, and it is a close relative of human T-cell leukemia virus type 1 (HTLV-1) and type 2. Both viral replications start with the reverse transcription of virion RNA into cDNA. The linear double-strand DNA (dsDNA) derived from the viral single strand RNA genome contains long terminal repeats (LTRs) found near both ends of the RNA and the dsDNA becomes integrated into the chromosomal DNA of the host to form a provirus, via a mechanism involving the viral integrase protein [1]. BLV is the causative agent of enzootic bovine leukosis (EBL), a malignant B-cell lymphoma [2]. Many cattle are asymptomatic carriers of this virus. Approximately 30% of BLV-infected cattle develop persistent lymphocytosis, and 0.1% to 5% of these develop EBL [2,3]. EBL is reportable to the World Organization for Animal Health (OIE), with disease incursion affecting international trade. EBL is also designated as a reportable disease by the Act on Domestic Animal Infectious Diseases Control in Japan, and any EBL-positive carcasses must be discarded when identified during meat inspection. In addition to causing EBL, BLV infection reduces milk production [4]. The annual economic loss in the United States from reduced milk production due to BLV infection is estimated to be USD 525 million [5]. Given that BLV appears to cause multiple immune system disruptions affecting both cellular and humoral immunity [6,7], animals infected with BLV are more susceptible to a number of pathogens, and infected cows are more susceptible to clinical mastitis [8], and are likely responsible for associations with decreased dairy production and decreased productive lifespan [9]. Furthermore, animals with BLV high proviral load were associated with carcass weight reduction [10].

BLV infection causes serious problems for the cattle industry. From 2009 to 2010, a nationwide survey of BLV in Japan revealed that the prevalence was 28.7% in beef breeding stock and 40.7% in dairy cattle [11]. The annual number of reported bovine leukosis (BL) cases was only 99 in 1998, but that number has increased rapidly and reached 4113 heads in 2019 [12]. BL has been classified into two different clinical syndromes, EBL and sporadic bovine leukosis (SBL). SBL has not been associated with any etiological agent, and approximately 99% of BL cases in Japan can be considered as EBL [13,14]. In 2015, the Ministry of Agriculture, Forestry, and Fisheries of Japan issued “Guidelines for Biosecurity Measures of Enzootic Bovine Leukosis” [15]. Eradication programs are currently being implemented, mainly by livestock hygiene centers in Japan’s prefectures.

Polymerase chain reaction (PCR), which detects the BLV proviral genome in peripheral blood mononuclear cells (PBMCs), has been used to rapidly, sensitively, and specifically detect BLV-infected cattle [2]. Recently, real-time quantitative PCR (qPCR), which can measure BLV provirus copy number—that is, the proviral load (PVL)—has been introduced worldwide. Because PVL in whole blood is correlated with the risk of horizontal and vertical transmission, its detection is used for the effective elimination of BLV-infected cattle on farms [16,17,18]. In addition, the PVL in biopsied lymph nodes can be used as a diagnostic index for predicting EBL [17]. In HTLV-1 infection, the PVL in PBMCs is strongly correlated with the severity of adult T-cell leukemia (ATL) [19,20]. Therefore, it is likely that the PVL and percentage of infected cells in whole blood could be used as an indicator of BLV disease progression and may be an important for producers and veterinarians to manage beef and dairy cattle production.

Since BLV infection in cattle is lifelong and gives rise to a persistent antibody response, serological tests are important as prescribed tests for animal trade [2]. However, there is no way of distinguishing passively transferred maternal antibodies from those resulting from active infection. Active infection can be confirmed by the detection of BLV provirus by the PCR. Several qPCR protocols for BLV provirus detection have been published [21,22,23,24]. The target regions for BLV provirus detection described in the OIE’s Terrestrial Manual are the envelope (env) region and the polymerase (pol) region, and detection of these regions are used by the international beef trade in many countries [2,25]. In Japan, three qPCR kits with different targets, chemistry, and reaction conditions are commercially available and are now routinely used for BLV provirus detection: CY415 [17] uses a chimeric cycling probe (cycleave) [26] targeting the tax region; RC202 uses a TaqMan probe [27] targeting the pol region; and CoCoMo [22] uses a TaqMan probe targeting the long terminal repeat (LTR) region. To interpret PVL as determined by the different qPCRs used in Japan, here we compared the CY415, RC202, and CoCoMo qPCRs by using whole-blood samples and tumor tissue samples derived from naturally BLV-infected cattle.

## 2. Materials and Methods

### 2.1. Animals

Whole-blood and serum samples were collected between 2016 and 2019 from 317 naturally BLV-infected cattle (165 Holstein–Friesian [HF] and 152 Japanese Black [JB]) bred on farms in the Tohoku region of Japan. The samples were sent to Iwate University for confirmation of diagnoses of BLV infection by local veterinarians (Appendix A). The study was based on voluntary participation in the EBL eradication program of the Agricultural Mutual Aid Association of the Tohoku area (NOSAI Tohoku). Consent for participation was obtained from the farm owners after a local veterinarian had explained the study.

Tumor tissue samples were collected from 32 cattle during 2018 through 2020 at a meat inspection center in the town of Shiwa, in Iwate Prefecture. Twenty-eight animals were diagnosed with BL (not differentiated into EBL or SBL) by the veterinarian at the meat inspection center, and the remaining four were diagnosed with EBL by PCR, ELISA, and pathological examination at Iwate University (Appendix A).

### 2.2. qPCR and ELISA

Genomic DNA was extracted from EDTA-treated whole blood with a magLEAD 12 gC nucleic-acid-extraction instrument (Precision System Science, Chiba, Japan). Genomic DNA from tumor tissues was extracted by using a DNeasy Blood and Tissue kit (Qiagen K.K., Tokyo, Japan) in accordance with the manufacturer’s instructions. The concentrations of DNAs obtained were calculated by the absorbance at a wavelength of 260 nm on a NanoDrop One spectrophotometer (Thermo Fisher Scientific K.K., Tokyo, Japan).

BLV PVLs were determined by using the three different commercial qPCR kits: CY415 (Takara Bio, Shiga, Japan), RC202 (Takara Bio), and CoCoMo (Riken Genesis, Tokyo, Japan). CY415 and RC202 testing was performed in accordance with the manufacturer’s instructions. CoCoMo testing was performed with premix Ex Taq (Probe qPCR; Takara Bio) in accordance with the manufacturer’s instructions. The qPCRs were performed on a QuantStudio 3 Real-Time PCR System (Thermo Fisher Scientific K.K.). Standard curves were generated by creating 10-fold serial dilutions of the standard plasmids included in the kits; the plasmids were amplified by using the appropriate PCR primers. The standards for calibration ranged from 100 to 105 copies/reaction and were run in duplicate. PVL was indicated as the copy number of BLV provirus per 10 ng DNA. The proportion of BLV-infected cells was calculated by using the following equation. (There were two copies of the ribonuclease [RNase] P gene per cell. The copy number of the RNase P gene was measured by using RC202 qPCR, which is a duplex qPCR and amplifies the RNase P and BLV pol genes.):Proportion of BLV-infected cells = [BLV tax or pol copy number or LTR copy number ÷ 2] ÷ (RNase P copy number ÷ 2) × 100]

Anti-BLV antibodies were determined by ELISA (Nippon Gene Co., Tokyo, Japan) in accordance with the manufacturer’s instructions.

### 2.3. DNA Sequencing

The PVLs determined by CoCoMo qPCR were lower than those determined by the other qPCRs, i.e., RC202 and CY415 in tumor samples L1, L5, and L7 (see Appendix A). Sequencing of DNA derived from three tumor tissue samples was performed by using an Illumina NovaSeq 6000 sequencing platform (Illumina, San Diego, CA, USA). The library was prepared by using a NEBNext Ultra DNA Library Prep Kit for Illumina (Cat. No. E7370, New England BioLabs, Ipswich, MA, USA). The prepared libraries were subjected to concentration measurement by Qubit 2.0 fluorometer (Life Technologies, Carlsbad, CA, USA), molecular length measurement by Agilent 2100 bioanalyzer (Agilent Technologies, Palo Alto, CA, USA), and q-PCR. Whole-genome sequencing was performed by Rhelixa Inc. (Tokyo, Japan). Adapter sequences and low-quality sequences were filtered out from the raw reads by using cutadapt version 1.18 [28] and Trimmomatic version 0.36 [29], respectively. All clean reads were assembled de novo with a k-value of 96 by using AbySS v2.3.3 [30]. The BLV proviral sequence was analyzed by using CLC genomics workbench software ver. 12 (Qiagen, Hilden, Germany).

### 2.4. Statistical Analysis

Differences between the BLV PVLs obtained by using the different qPCR reagents (CY415, RC202, and CoCoMo) were assessed by using a Kruskal–Wallis test with ad hoc Bonferroni’s multiple comparison. Correlations between the PVLs obtained with the different reagents were assessed by the Spearman’s correlation coefficients. The proportions of BLV-infected cells were also analyzed in the same manner. These data analyses were performed by using R, a language and environment for statistical computing (R Core Team, 2020; URL https://www.R-project.org, accessed on 10 February 2022), and statistical significance was determined as *p* < 0.05.

## 3. Results

### 3.1. Sensitivity of the Three qPCRs

ELISA analyses showed that the sera from all 317 whole-blood samples were positive for anti-BLV antibody (data not shown). Therefore, all of the cattle were BLV infected. The sensitivities (referring to the presence of anti-BLV antibody detected by ELISA) were, for CY415, 98.4% (=312/317); RC202, 99.4% (=315/317); and CoCoMo, 99.4% (=315/317). No PVL was detected in two whole-blood samples from HF cattle (animal numbers 7 and 136; Appendix A) by any qPCR and in three whole-blood samples from HF cattle (animal numbers 138, 283, and 293; Appendix A) by CY415.

### 3.2. PVL and the Proportion of BLV-Infected Cells in Whole-Blood Samples by Different qPCRs

We compiled the results of descriptive analyses of PVL in whole-blood samples; they were not normally distributed (*p* < 0.05, Shapiro–Wilk normality test) (Table 1 and Figure 1A). The median (and interquartile range) of the PVL were, for CY415, 119.3 (15.9, 347.0); for RC202, 372.2 (55.1, 1037.6); and for CoCoMo, 658.4 (124.1, 1681.5). These differences were statistically significant (*p* < 0.05, Kruskal-Wallis test). PVL determined by CoCoMo or by RC202 was significantly higher than that by CY415 (*p* < 0.05, Bonferroni’s multiple comparison). There were strong correlations between the PVLs by each qPCR (>0.8; Figure 1B–D; Spearman’s correlation coefficient, *p* < 0.0001).

We compiled the results of descriptive analyses of the proportions of BLV-infected cells in whole-blood samples; they were not normally distributed (*p* < 0.05, Shapiro-Wilk normality test) (Table 1 and Figure 2A). The median (and interquartile range) of the proportion of BLV-infected cells was, for CY415, 2.6 (0.03, 9.5); for RC202, 9.9 (2.1, 26.5); and for CoCoMo, 16.9 (4.7, 43.5). These differences were statistically significant (*p* < 0.05, Kruskal–Wallis test). The proportion of BLV-infected cells determined by CoCoMo was significantly higher than that by CY415 or RC202, and the proportion of BLV-infected cells determined by RC202 was significantly higher than that determined by CY415 (*p* < 0.05, Bonferroni’s multiple comparison). There were strong correlations between PVL and the proportion of BLV-infected cells in each qPCR (>0.95; Figure 2B–D, Spearman’s correlation coefficient, *p* < 0.0001).

PVL and the proportion of BLV-infected cells were compared between HF and JB cattle. PVL in whole-blood samples was not significantly different between HF and JB cattle when CY415 was used (*p* = 0.203, Mann–Whitney test). In contrast, the PVLs in whole-blood samples from HF cattle were significantly higher than those from JB cattle when RC202 or CoCoMo was used (*p* < 0.001 and *p* < 0.0001 respectively; Mann–Whitney test, Figure 3A). The proportion of BLV-infected cells in whole-blood samples from HF cattle was significantly higher than that from JB cattle when any of the three qPCRs was used (*p* < 0.05 for CY415, *p* < 0.0001 for RC202, and *p* < 0.0001 for CoCo Mo; Mann–Whitney test, Figure 3B).

### 3.3. PVL and the Proportion of BLV-Infected Cells in Tumor Tissue Samples by Different qPCRs

We compiled the results of descriptive analyses of PVL in the tumor tissue samples; the results were not normally distributed (*p* < 0.05, Shapiro–Wilk normality test) (Table 2 and Figure 4A). The median (and interquartile range) of PVL were, for CY415, 1016.2 (345.6, 2152.8); for RC202, 2255.7 (712.2, 4362.7); and for CoCoMo, 2793.1 (501.6, 6817.5). These differences were statistically significant (*p* < 0.05, Kruskal–Wallis test). PVL determined by CoCoMo or by RC202 was significantly higher than that by CY415 (*p* < 0.0001, Bonferroni’s multiple comparison). There were strong correlations between the PVLs determined by each qPCR, (>0.8; Figure 4B–D; Spearman’s correlation coefficient, *p* < 0.05).

We then compiled the results of descriptive analyses of the proportions of BLV-infected cells in the tumor tissue samples; the results were not normally distributed (*p* < 0.05, Shapiro–Wilk normality test) (Table 2 and Figure 5A). The median (and interquartile range) of the proportion of BLV-infected cells were, for CY415, 45.0 (30.4, 76.8); for RC202, 99.0 (67.6, 147.3); and for CoCoMo, 122.4 (48.8, 202.7). These differences were statistically significant (*p* < 0.05, Kruskal–Wallis test). The proportion of BLV-infected cells determined by CoCoMo or by RC202 was significantly higher than that determined by CY415 (*p* < 0.001, Bonferroni’s multiple comparison). There were weak to strong correlations between PVL and the proportion of BLV-infected cells determined by each qPCR (0.45 to 0.8; Figure 5B–D; Spearman’s correlation coefficient, *p* < 0.05)

### 3.4. Nucleotide Sequences of BLV Proviruses in Tumor Tissue Samples

As described above, the PVLs determined by CoCoMo or RC202 were significantly higher than those determined by CY415. The PVLs determined by CY415 were far lower than those determined by the other qPCRs in tumor samples L17 and L26 (Appendix A). In contrast, however, the PVLs determined by CoCoMo were lower than those determined by the other qPCRs in tumor samples L1, L5, and L7 (Appendix A). To investigate the reason for the latter differences, we determined the nucleotide sequences of BLV proviruses in these tumor tissue samples. We found that L1 and L7 had a defective 5ʹ-LTR and L5 had a mutation at the binding site for a CoCoMo primer in the 5ʹ-LTR region (data not shown).

## 4. Discussion

By using whole-blood and tumor tissue samples from naturally BLV-infected cattle, we compared three commercial BLV qPCR used in Japan, namely the cycleave-based CY415 targeting the tax region; the TaqMan probe-based RC202 targeting the pol region; and the TaqMan probe-based CoCoMo targeting the LTR region. The sensitivity of each qPCR was approximately 99%. However, several points should be taken into consideration when these qPCRs are to be used for the diagnosis and prognosis of BLV infection.

We found that the PVL and the proportion of BLV-infected cells determined by the TaqMan probe-based RC202 or CoCoMo were significantly higher than those determined by the cycleave-based CY415. The reasons for these differences are not clear. However, the chemistry used for each qPCR may have contributed. The TaqMan PCR assay, first described by Holland et al. [27], uses the 5′-nuclease activity of Taq DNA polymerase. A fluorogenic probe, which specifically binds between the two PCR primers, is degraded in each PCR cycle. This degradation releases the fluorophore and thereby eliminates the quenching of the two fluorescence markers; amplification of the target gene is thus monitored as an increasing fluorescence signal. Cycleave PCR assay was first reported by Duck et al. [26]. A key component of this method is the sequence-specific DNA-RNA chimeric probe, which hybridizes to a complementary target DNA sequence. If the probe successfully hybridizes to its specific target sequence, the RNA strand of the RNA-DNA hybrid is degraded by RNase H and the probe fluoresces. Although the relative ease of designing sequence-specific probes makes cycling probe technology highly amenable to the detection of single-nucleotide polymorphisms, the higher sequence specificity of this technology may be the reason why its performance is lower than that of TaqMan-based qPCR.

We found strong correlations between the BLV PVL and the proportion of BLV-infected blood cells detected by each qPCR. In HTLV-1 infection in humans, the PVL in PBMCs is strongly correlated with the disease severity of ATL. PVL per 100 PBMCs—that is, the percentage of % HTLV-infected cells—has been used as a biomarker of ATL disease prognosis [19,20]. HTLV-infected patients with PVLs ranging from 4.2 to 28.6 copies/100 PBMCs (i.e., 4.2% to 28.6% of HTLV-infected cells) developed ATL, whereas those with PVLs lower than approximately 4 copies/100 PBMCs (4% of HTLV-infected cells) did not develop ATL [20]. Similar studies are needed in BLV-infected cattle to establish the relationship between the proportion of BLV-infected cells in the peripheral blood and EBL onset. Clearly, RC202 and CoCoMo are more sensitive than CY415 and should be used for such future investigations.

We found that the PVLs in whole-blood samples did not differ significantly between HF and JB cattle when CY415 was used. In contrast, the PVLs in whole-blood samples from HF cattle were significantly higher than those from JB cattle when RC202 or CoCoMo was used. These results emphasize that the choice of qPCR clearly influences the results of PVL analysis in BLV-infected cattle. The reason why the PVLs in HF cattle were higher than those in JB cattle is unclear. It might be because lymphocyte counts in HF cattle are inherently higher than those in JB cattle [31,32,33].

There were strong correlations among the three qPCR kits in the detection of PVL when DNA from tumor tissues was used (Figure 4B–D). These results were the same as those obtained by using peripheral blood DNA. Previously, we reported that cattle harboring more than 1000 BLV copies per 10 ng DNA in biopsy-derived tissue could be diagnosed with EBL [34]. Here (Appendix A), we obtained data similar to those in our previous study. The percentage of BLV-infected cells determined on the basis of RNase P-gene copies was also investigated by using RC202 (Appendix A). Mean cell infection rates and SD were, for CY415, 53% ± 36%; for RC202, 117% ± 78%; and for CoCoMo,139% ± 104%. These differences in cell infection rates seem reasonable given the low BLV detectability of CY415 compared with those of the two TaqMan chemistry qPCRs. Interestingly, the percentage of HTLV-1-infected cells in ATL has been reported as 117% ± 51% [35]. Our EBL data are therefore in very good agreement with those for ATL; especially when RC202 was used, our data were very close to those in the ATL study [35]. Adding a diagnostic marker, such as the percentage of infected cells, would allow us to make a more accurate diagnosis of EBL.

We found defects or a mutation in the 5ʹ-LTR region of BLV provirus in three of the 32 tumor tissue samples; the PVLs determined by CoCoMo were therefore lower than those determined by the other qPCRs. These results suggest that mutations or defects in the qPCR target regions are not rare in tumors in BLV-infected cattle and potentially hamper PVL analysis by using qPCR. It is interesting to note that we also found that the PVL determined by CY415 was much lower than those determined by the other qPCRs in two tumor tissue samples, L17 and L26 (Appendix A). The implication was that this was due to a mutation in the tax region binding a CY415 primer. However, it is unclear whether or not this was the case, because the nucleotide sequences of the CY415 primers have not been disclosed. In HTLV-1 infection in humans, defects in the 5ʹ-LTR region have been observed in approximately 40% of patients with aggressive-type ATL and 6% of patients with chronic-type ATL, suggesting that HTLV-1-infected cells with 5ʹ-LTR defects tend to become leukemic [36]. In addition, nonsense mutations or defects are often seen in the tax region in HTLV-1-infected leukemic cell lines and fresh ATL cells [37]. However, it is largely unclear what mutations or defects are contained in which regions of the BLV provirus, and at what frequency in the tumors of BLV-infected cattle.

In conclusion, our results suggest that the TaqMan-based RC202 or CoCoMo is better than cycleave-based CY415 for PVL analysis in BLV-infected cattle in the future. Because the pol region is recommended by OIE as the target for qPCR and is used worldwide [24,25], RC202 may be convenient at the moment for the global cattle trade. However, to select a suitable qPCR for the diagnosis and prognosis of BLV infection in cattle, further studies are needed to determine which region of the BLV provirus possesses the least mutations or defects in tumors.

## Figures and Tables

**Figure 1 viruses-14-01182-f001:**
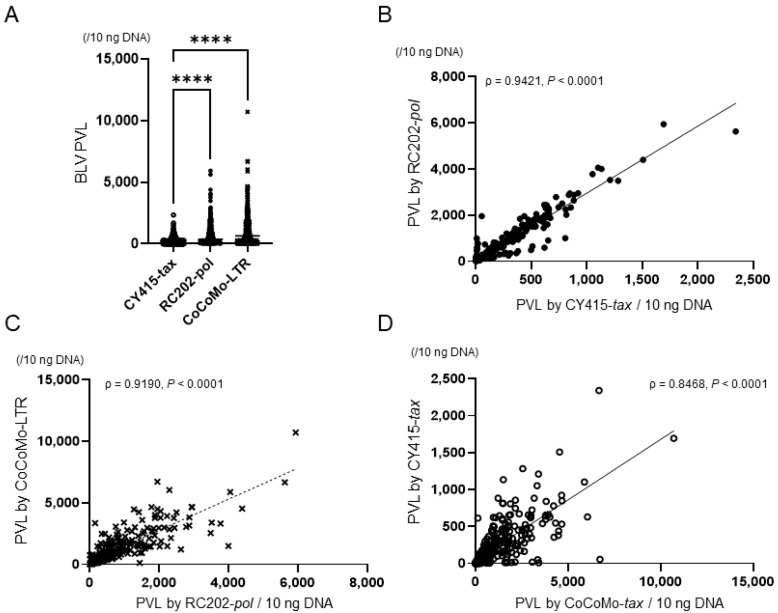
Bovine leukemia virus (BLV) proviral load (PVL) in whole-blood samples, as determined by different qPCRs. (**A**) BLV PVL. Differences between values obtained were assessed by using Kruskal-Wallis tests with Bonferroni’s multiple comparison as an ad hoc test. Horizontal bars indicate median values. ****, *p* < 0.0001. (**B**–**D**) Correlations between PVLs determined by each qPCR. Spearman’s correlation coefficient (ρ) and the levels of significance (P) regarding values are indicated in the graphs. Diagonal lines are regression lines.

**Figure 2 viruses-14-01182-f002:**
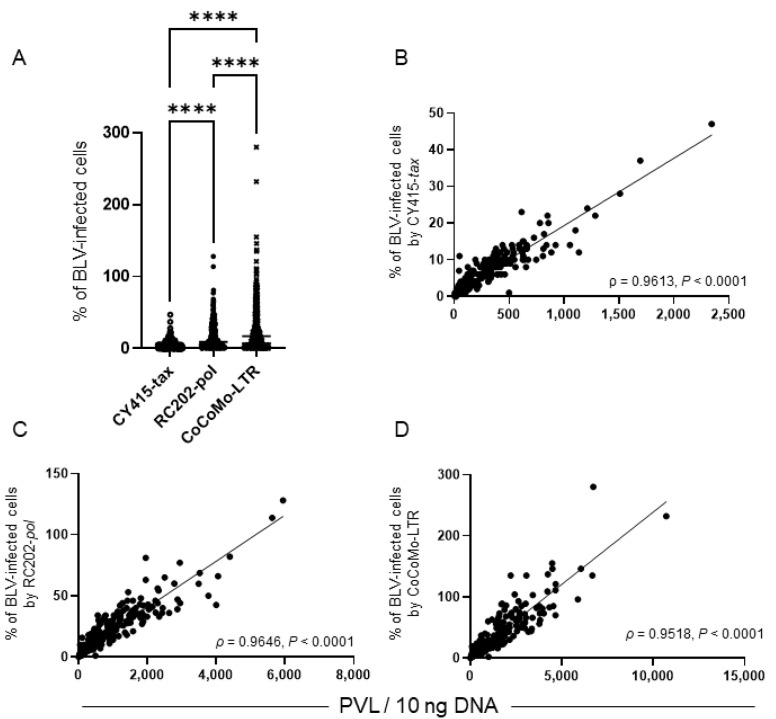
Proportions of bovine leukemia virus (BLV)-infected cells in whole-blood samples, as determined by different qPCRs. (**A**) Proportion of BLV-infected cells. Differences between values obtained were assessed by using Kruskal-Wallis tests with Bonferroni’s multiple comparison as an ad hoc test. Horizontal bars indicate median values. ****, *p* < 0.0001). (**B**–**D**) Correlations between proviral load (PVL) and the proportion of BLV-infected cells determined by each qPCR. Spearman’s correlation coefficient (ρ) and levels of significance (P) regarding values are indicated in the graphs. Diagonal lines are regression lines.

**Figure 3 viruses-14-01182-f003:**
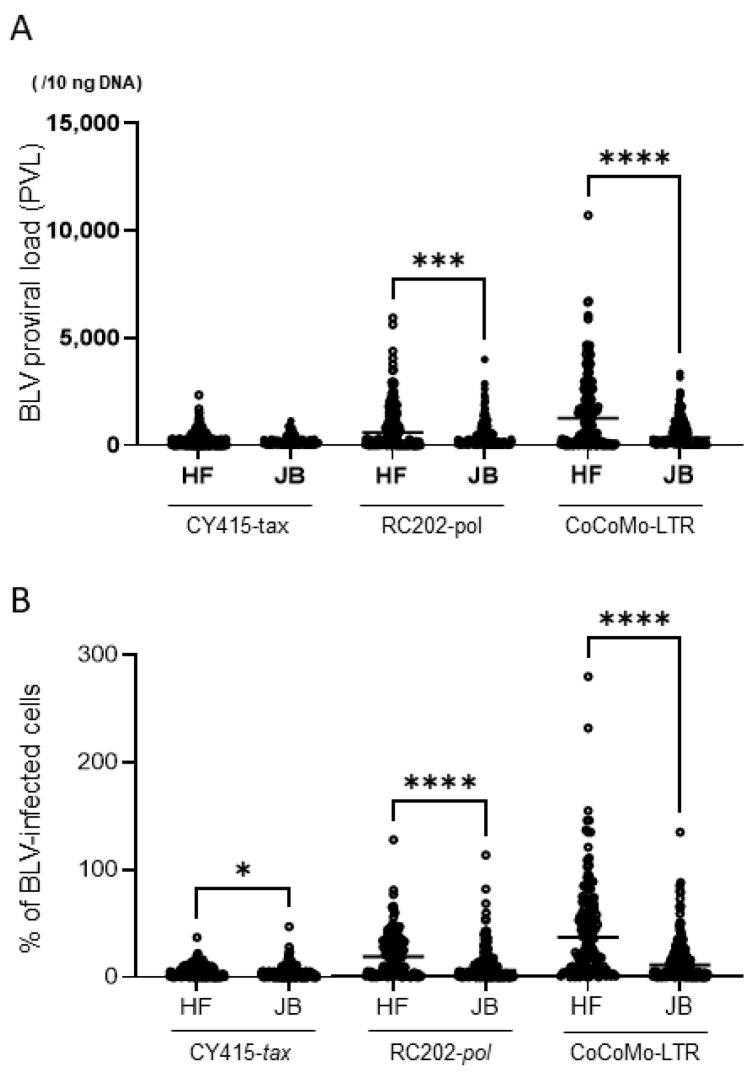
Comparison of proviral load (PVL) and the proportion of bovine leukemia virus (BLV)-infected cells between Holstein–Friesian (HF) cattle and Japanese Black (JB) cattle. (**A**) BLV PVL determined by different qPCRs. (**B**) Proportion of BLV-infected cells determined by different qPCRs. Differences between values obtained were assessed by using the Mann–Whitney U-test (***, *p* < 0.001; ****; *p* < 0.0001).

**Figure 4 viruses-14-01182-f004:**
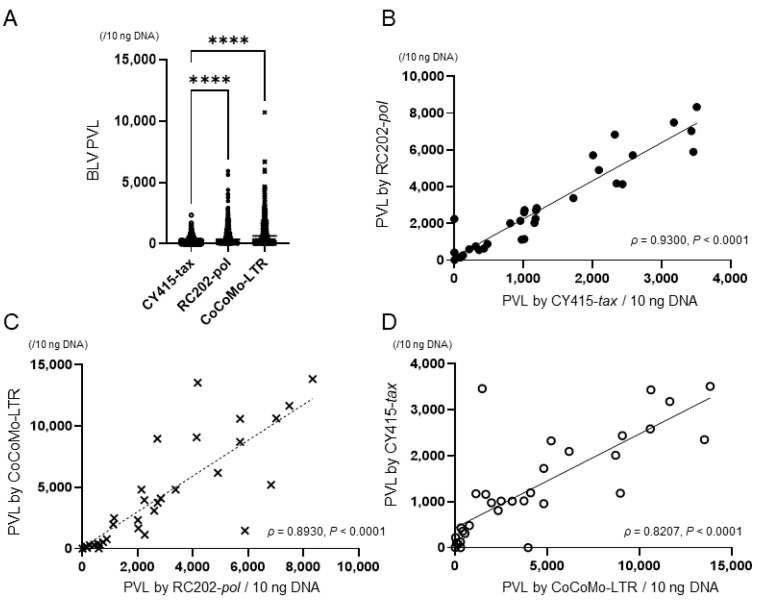
Bovine leukemia virus (BLV) proviral load (PVL) in tumor tissue samples, as determined by different qPCRs. (**A**) BLV PVL. Differences between values obtained were assessed by using Kruskal–Wallis tests with Bonferroni’s multiple comparison as an ad hoc test. Horizontal bars indicate median values. ****, *p* < 0.0001. (**B**–**D**) Correlations between PVLs determined by each qPCR. Spearman’s correlation coefficient (ρ) and levels of significance (P) regarding values are indicated in the graphs. Diagonal lines are regression lines.

**Figure 5 viruses-14-01182-f005:**
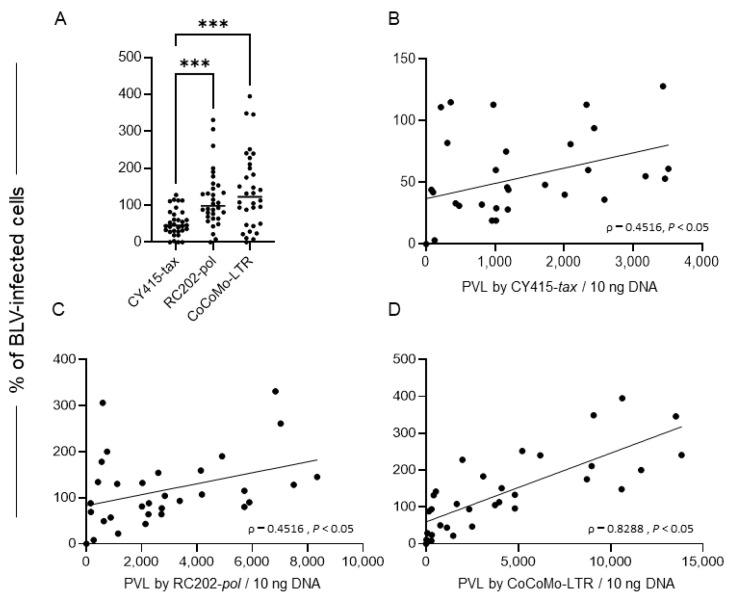
Proportions of bovine leukemia virus (BLV)-infected cells in tumor tissue samples, as determined by different qPCRs. (**A**) Proportion of BLV-infected cells. Differences between values obtained were assessed by using Kruskal–Wallis tests with Bonferroni’s multiple comparison as an ad hoc test. Horizontal bars indicate median values. ***, *p* < 0.001. (**B**–**D**) Correlation between proviral load (PVL) and the proportion of BLV-infected cells determined by each qPCR. Spearman’s correlation coefficient (ρ) and levels of significance (P) regarding values are indicated in the graphs. Diagonal lines are regression lines.

**Table 1 viruses-14-01182-t001:** Descriptive statistics for bovine leukemia virus (BLV) proviral load (PVL) and the proportable 312. blood samples, as determined by using three different qPCR.

Type of Reagent	0 Percentile (Minimum)	25 Percentile	50 Percentile (Median)	75 Percentile	100 Percentile (Maximum)	Shapiro–Wilk Normality Test *p*
BLV PVL						
CY415-*tax*	0.1	15.9	119.3	347	2343.7	<0.05
RC202-*pol*	0.2	55.1	372.3	1037.6	5943.2	<0.05
CoCoMo-LTR	0.2	124.1	658.4	1681.5	10713	<0.05
Proportion of BLV-infected cells						
CY415-*tax*	<0.0001	0.03	2.6	9.5	50	<0.05
RC202-*pol*	0.004	2.1	9.9	26.5	128.5	<0.05
CoCoMo-LTR	0.003	4.7	16.9	43.5	280.4	<0.05

**Table 2 viruses-14-01182-t002:** Descriptive statistics for bovine leukemia virus (BLV) proviral load (PVL) and the proportions (%) of BLV-infected cells from 32 tumor samples, as determined by using the three different qPCRs.

Type of Reagent	0 Percentile (Minimum)	25 Percentile	50 Percentile (Median)	75 Percentile	100 Percentile (Maximum)	Shapiro–Wilk Normality Test *p*
BLV PVL						
CY415-*tax*	0.18	345.6	1016.2	2152.8	3511.6	<0.05
RC202-*pol*	10.1	712.2	2255.7	4362.7	8338.9	<0.05
CoCoMo-LTR	6.8	501.6	2793.1	6817.5	13845.1	<0.05
Proportion of BLV-infected cells (%)						
CY415-*tax*	0.006	30.4	45	76.8	127.5	<0.05
RC202-*pol*	0.3	67.6	99	147.3	331.2	<0.05
CoCoMo-LTR	0.2	48.8	122.4	202.7	394.8	<0.05

## Data Availability

All the data generated for this publication have been included in the current manuscript.

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
