# Peer review of "Comparative Evaluation of Three Commercial Quantitative Real-Time PCRs Used in Japan for Bovine Leukemia Virus"

_viruses, 2022, doi:10.3390/v14061182_

Round 1
Reviewer 1 Report
viruses-1692957
This manuscript reports the comparison of three different commercially available BLV tests using 317 whole blood samples and 32 tumor samples collected from cattle that had previously been diagnosed as positive for BLV. The three tests targeted three different genomic regions,
tax region (CY415), pol region (RC202) and the long terminal repeat region (CoCoMo). The OIE manual describes a test for BLV based on detection of the pol region. All three tests had greater than 98% accuracy, with the test based on the long terminal repeat region having the lowest accuracy. The authors discussed, at length, the percentage of infected cells detected by each test. While the percentage of infected cells may be correlated with progression of disease, this correlation has not been demonstrated in cattle and it is not clear if such tests are typically run by diagnosticians. It also appears that percentage of infected cells may be a function of breed. The authors need to clearly state, for the readers benefit, why a producer or veterinarian would be interested in percentage of infected cells. Does the percentage of infected cells impact on production management or is this a metric that, at this time, is only of importance to researchers? As the major portion of the manuscript deals with these results, it is important for the authors to explain why this metric so important.
Some general comments;
Don’t use “% tile” instead use percentile
Why did the authors select the three tumor samples for sequencing? What were the reasons for sequencing the virus from these three particular samples?
Who are the “local veterinarians” (line 90) that diagnosed the initial cases? Are they trained in BLV diagnosis?
Avoid terms like “precisely detected”, instead use the term detected.
Avoid terms like “comparatively evaluated”, instead use the term compared
Author Response
Comments to Reviewer: 1
This manuscript reports the comparison of three different commercially available BLV tests using 317 whole blood samples and 32 tumor samples collected from cattle that had previously been diagnosed as positive for BLV. The three tests targeted three different genomic regions, tax region (CY415), pol region (RC202) and the long terminal repeat region (CoCoMo). The OIE manual describes a test for BLV based on detection of the pol region. All three tests had greater than 98% accuracy, with the test based on the long terminal repeat region having the lowest accuracy. The authors discussed, at length, the percentage of infected cells detected by each test. While the percentage of infected cells may be correlated with progression of disease, this correlation has not been demonstrated in cattle and it is not clear if such tests are typically run by diagnosticians. It also appears that percentage of infected cells may be a function of breed. The authors need to clearly state, for the readers benefit, why a producer or veterinarian would be interested in percentage of infected cells. Does the percentage of infected cells impact on production management or is this a metric that, at this time, is only of importance to researchers? As the major portion of the manuscript deals with these results, it is important for the authors to explain why this metric so important.
We modified and added the sentences (L59-61 and L81-83) in the introduction that at present PVL is mainly used as an indicator of EBL and productivity loss, but that in the future the percentage of infected cells will be an important indicator not only for researchers but also for production management.
Some general comments;
Don’t use “% tile” instead use percentile
We modified “% tile” to “percentile” in table 1 and 2.
Why did the authors select the three tumor samples for sequencing? What were the reasons for sequencing the virus from these three particular samples?
Our current results showed that PVLs determined by RC202-pol or CoCoMo-LTR were generally higher than those determined by CY415-tax. In contrast, three tumor samples, L1, L5 and L7 of 32 tumor tissue samples showed lower PVLs determined by CoCoMo-LTR than those determined by RC202-pol or CY415-tax. Therefore, we selected these three tumor samples for sequencing to investigate LTR defects. We described the reason why the three tumor samples were selected in 2.3 in Materials and methods of revised manuscript.
Who are the “local veterinarians” (line 90) that diagnosed the initial cases? Are they trained in BLV diagnosis?
All local veterinarians in this study belong to the Agricultural Mutual Aid Association of the Tohoku area (NOSAI Tohoku). They are clinical veterinarians for cattle and are well trained in the clinical features of EBL and the diagnosis of BLV by blood tests.
Avoid terms like “precisely detected”, instead use the term detected.
we modified the word with “precise” according to reviewer’s suggestion as below.
- "precisely interpret PVL” to “interpret PVL” in L24, L35 and L95
- “precisely PVL analysis” to “PVL analysis” in L345.
Avoid terms like “comparatively evaluated”, instead use the term compared
We modified “comparatively evaluated” to “compared” in L25, L96 and L286.
Reviewer 2 Report
The paper submitted for review under the title “Comparative Evaluation of Three Commercial Quantitative Real-time PCRs used in Japan for Bovine Leukemia Virus ” by Yoneyama S. et al. is a diligently conducted new study on the comparison of qPCR diagnostic kits for detection of proviral DNA BLV, commercially available in Japan. The manuscript is clearly written and technically sound. This type of analysis is essential to permit accurate diagnosis of infection with the bovine leukemia virus. In general, the paper is well written and easy to follow. The methods are appropriate and adequately conducted.
Specific comments:
The “Introduction” should be supplemented with information clarifying the nature of proviral DNA, since it cannot be assumed that all readers are familiar with the replication mechanism of the bovine leukaemia virus
In the line #65 the high specificity of qPCR should be mentioned
Line #76 - This sentence may be modified according to the literature cited. In fact, it appears that the jury is still out on this point.
In the “Introduction” the authors should further point out the relevant role of serological tests in the diagnosis of BLV infections and their importance as a prescribed tests for animal trade, according to OIE recommendations
Lines #123-124 - since sera were tested by ELISA (as I understand it in these studies) the relevant information about serum samples should be incorporated in the section 2.1
Lines #227-128 - this sentence is unclear, and I don’t understand the use of “amount of LTR sequences ” in this context.
Why only the samples from three animals were selected for sequencing? Perhaps a higher number of samples sequenced could provide more data to explain the relatively lower sensitivity of CoCOMo and Cy415 qPCRs
Author Response
Comments to Reviewer: 2
The “Introduction” should be supplemented with information clarifying the nature of proviral DNA, since it cannot be assumed that all readers are familiar with the replication mechanism of the bovine leukaemia virus
We added the sentences about the nature of proviral DNA in L41-45.
In the line #65 the high specificity of qPCR should be mentioned
We modified the sentence as follows; “Polymerase chain reaction (PCR), which detects the BLV proviral genome in peripheral blood mononuclear cells (PBMCs), has been used to rapidly, sensitively and specifically detect BLV-infected cattle” in L71–73.
Line #76 - This sentence may be modified according to the literature cited. In fact, it appears that the jury is still out on this point.
We modified the sentences as follws; “The target regions for BLV provirus detection described in the OIE’s Terrestrial Manual are the envelope (env) region for nested PCR and the polymerase (pol) region for real-time PCR, and detection of these regions are used by the international beef trade in many countries” in L88-90.
In the “Introduction” the authors should further point out the relevant role of serological tests in the diagnosis of BLV infections and their importance as a prescribed tests for animal trade, according to OIE recommendations
We added the sentences for serological test as follows; “Since infection with BLV in cattle is lifelong and gives rise to a persistent antibody response, serological tests are importance as a prescribed tests for animal trade. However, there is no way of distinguishing passively transferred maternal antibodies from those resulting from active infection. Active infection, however, can be confirmed by the detection of BLV provirus by the PCR.” in L83-87.
Lines #123-124 - since sera were tested by ELISA (as I understand it in these studies) the relevant information about serum samples should be incorporated in the section 2.1
We modified the sentence in L101.
Lines #127-128 - this sentence is unclear, and I don’t understand the use of “amount of LTR sequences ” in this context.
In the revised manuscript, we modified the sentence there in L141-145, so the part the reviewer pointed out is removed.
Why only the samples from three animals were selected for sequencing? Perhaps a higher number of samples sequenced could provide more data to explain the relatively lower sensitivity of CoCOMo and Cy415 qPCRs
Our current results showed that PVLs determined by RC202-pol or CoCoMo-LTR were generally higher than those determined by CY415-tax. In contrast, three tumor samples, L1, L5 and L7 of 32 tumor tissue samples showed lower PVLs determined by CoCoMo-LTR than those determined by RC202-pol or CY415-tax. Therefore, we selected these three tumor samples for sequencing to investigate LTR defects. We described the reason why the three tumor samples were selected in 2.3 in Materials and methods of revised manuscript.
We appreciated reviewer’s advice. We would like to sequence more samples to explain the relative low sensitivity of CoCOMo and Cy415 qPCR in future.
Other
・The sentence of L167-169 in the previous manuscript has been removed, because the sentence was instruction of the word template.